# Promiscuity in Polyphenol–Protein Interactions—Monitoring Protein Conformational Change upon Polyphenol–Protein Binding by Nano-Differential Fluorimetry (Nano-DSF)

**DOI:** 10.3390/molecules30040965

**Published:** 2025-02-19

**Authors:** Dorothea Schmidt, Amelie Wohlers, Nikolai Kuhnert

**Affiliations:** School of Science, Constructor University, 28759 Bremen, Germany; dschmidt@constructor.university (D.S.); awohlers@constructor.university (A.W.)

**Keywords:** polyphenols, protein, polyphenol–protein binding, differential fluorimetry, protein conformation, diet

## Abstract

In this article, we introduce nano-differential fluorimetry (nano-DSF) as an analytical technique that is suitable for investigating polyphenol–protein interactions in solution. Nano-DSF monitors conformational changes in proteins induced by external agents upon interaction at the molecular level. We demonstrate the suitability of this technique to qualitatively monitor an interaction between selected dietary polyphenols and selected proteins including BSA, ovalbumin, amylase, pepsin, trypsin, mucin and ACE-1. Protein conformational changes induced by dietary polyphenols can be investigated. As a major advantage, measurements are carried out at a high dilution, avoiding the precipitation of polyphenol–protein complexes, allowing the rapid and efficient acquisition of quantitative and qualitative binding data. From this concentration, quantitative binding data could be obtained from the fluorescence response curve in line with published values for the association constants. We demonstrate that qualitative interactions can also be established for real food extracts such as cocoa, tea or coffee containing mixtures of dietary polyphenols. Most importantly, we demonstrate that polyphenols of very different structural classes interact with the same protein target. Conversely, multiple protein targets show an affinity to a series of structurally diverse polyphenols, therefore suggesting a dual level of promiscuity with respect to the protein target and polyphenol structure.

## 1. Introduction

Polyphenols (PPs) are ubiquitous plant secondary metabolites and hence are an integral part of the human diet. Current dietary advice suggests the consumption of a diet rich in plant material, advice based on numerous epidemiological studies and clinical intervention studies, based on the assumption of a beneficial role of polyphenols in human health [1,2].

Selected foods containing high levels of polyphenols have been shown to possess beneficial health effects such as improving cardiovascular health, reducing inflammatory processes, acting as a chemopreventive against cancer, improving mental performance and many more [3,4].

It is worth pointing out that such beneficial health effects can be divided into effects where overwhelming evidence results in a broad scientific consensus such as coffee preventing diabetes [5], whereas on most occasions beneficial health effects are based on contradictory evidence and should be considered as controversial. The mechanism of action by which polyphenols improve human health is still not clearly understood. For a long time the “anti-oxidant hypothesis” dominated the field; recent consensus, however, points in other directions, which has bene discussed in detail in several recent reviews [6,7].

Due to the numerous effects observed from a polyphenol-rich diet, it must be assumed that a multitude of biological targets exist that bind to polyphenols of various structures and thus modulate human biochemical and signal pathways. As an example, it is a worthwhile exercise to count the number of biological targets that interact with highly abundant and common polyphenolic compounds such as EGCG or 5-caffeoyl quinic acid [8]. It can be further assumed that a majority of these biological targets are proteins and fundamental studies on polyphenol–protein interactions will provide important insights into many open questions.

Due to their aromatic and polyhydroxylated structure, polyphenols are able to interact with proteins, forming multiple hydrogen bonds via their OH moieties [9], accepting hydrogen bonds from protein amide NHs and additionally forming π–cation interactions with lysine and arginine residues as well as π–π interactions with aromatic protein side chains [10]. Indeed, polyphenols might be viewed as “protein glue”, a property exploited in many applications [11] including leather tanning, the removal of proteins from beverages or vice versa in the removal of polyphenols from beverages by protein-like materials, where PP–protein complexes with low solubility are formed. A good theoretical understanding of protein precipitation has been developed and key structural features in polyphenol identified [12,13]. In the human mouth polyphenols interact with salivary proteins to precipitate salivary protein, leading to the sensation of astringency [14].

This particular phenomenon of forming poorly soluble PP–protein complexes impedes analysis, in particular, quantitative investigations of PP–protein interactions since any method employed needs to operate within the solubility range of all three components: PPs, proteins and PP–protein complexes. In particular, the solubility products of protein–polyphenol complexes are typically low [15,16], sometimes in the µM range resulting in complex precipitation and thus evading solution-based analytical techniques [15]. Consequently, only on rare occasions have binding constants of this equilibrium been accessed experimentally, quantified and reported. More information is available on the enzyme inhibition constants of PPs as enzyme inhibitors. Such experiments do not necessarily require high PP concentrations that lead to complex precipitation. When surveying quantitative enzyme inhibition data available in the literature, we noted that in the large majority of cases, Hill coefficients of around five were observed. This leads us to the conclusion that for enzyme–PP interactions, multiple PPs bind to a given target, altering its overall conformation, which is associated with a loss of enzyme function. This mechanism resembles a non-selective allosteric inhibition mechanism. Only on rare occasions have Hill coefficients of one been reported, pointing towards selective binding at the enzyme active site [8].

Varying binding stoichiometries are next to solubility issues, a second challenge impeding the determination of quantitative binding data, since only after experimental determination of a correct stoichiometry can a meaningful mathematical model for curve fitting and hence estimation of equilibrium constants be realized. Such experimental stoichiometry determination requires relatively high PP and protein concentrations as data points are impeded by solubility issues [17].

In this article, we introduce nano-differential fluorimetry (nano-DSF) as a highly useful and promising tool to study PP–protein interactions both qualitatively and quantitatively [18]. Nano-DSF has been developed as a tool to study conformational changes in therapeutic proteins produced industrially by the “biologicals industry” [19]. Following protein expression and purification, the correct folding of the biological is of primary importance for its safe clinical use, as assessed by nano-DSF. Nano-DSF takes advantage of fluorescence changes in tryptophan (trp) and tyrosine (tyr) residues in proteins. It is well established that following absorption at 295 nm, tryptophan fluorescence crucially depends on the dielectric constant of its intermediate environment. Hence, tryptophan moieties buried within the lipophilic core of a protein show emission at 320 nm, whereas tryptophan moieties exposed to the aqueous environment surrounding a given protein in solution show an emission maximum of 350 nm. Hence, monitoring the difference in intensity between emissions at 330 and 350 nm provides a direct measure for conformational integrity or conformational change upon binding within a given protein, operating at low concentrations, thus avoiding insoluble complex formation. Therefore, this technique allows the investigation of protein conformational change induced by external agents such as PPs. Denaturation of proteins by other external agents such as urea has been determined by nano-DSF previously [20]. The sensitivity of the method allows additional measurements at concentration ranges much below the solubility product of PP–protein complexes, solving a key challenge already mentioned. We have recently introduced nano-DSF as a technique that is able to investigate the interaction between the human ACE-2 receptor and COVID spike protein and selected dietary polyphenols [21]. We now report on a detailed study of the technique including a larger set of examples both with respect to the proteins and polyphenols investigated.

## 2. Results

This article is arranged in three sections. Firstly, we introduce the nano-DSF method and demonstrate its suitability to determine PP–protein interactions. We show that three distinct types of concentration–fluorescence qualitative behaviors are observed depending on the nature of the protein and the PP under investigation. For selected examples, we have obtained quantitative binding data through curve fitting. Here, we validate the method by comparison to data available in the literature, in particular, binding data to serum bovine albumin. For our study we used fourteen selected PPs of dietary importance: caffeic acid, 5-caffeoyl quinic acid and 3,4 dicaffeoyl quinic acid, abundant in coffee; EGCG, which is abundant in green tea; pentagalloylglucose (PGG) in both anomeric configurations, which are found in pomegranate and mango; quercetin and its glycoside, which are abundant in onion and apple; malvidin and cyanidin glucosides and caftaric acid, which are abundant in grapes; and tannic acid as a mixture of galloyl glucoses. Chemical structures are given in Figure 1. On the protein side, we opted to study bovine serum albumin (BSA) due to the available reference data, ovalbumin and casein typical proteins relevant to the human diet, along with digestive enzymes abundant in the digestive tract such as amylase, lipase, pepsin, chemotrypsin and trypsin [22]. Finally, we added mucin, a protein secreted by goblet cells in the epithelial membrane in the digestive tract, and angiotensin-converting enzyme 1 (ACE-1) as a typical protein receptor with relevance to human health due to its role in blood pressure control. Interactions between dietary PPs and digestive enzymes might hold the key to understanding their effect in metabolic syndrome [23,24].

In the second part, we survey these various PP–protein combinations and qualitatively note down experimentally observed interactions. In the third section, we demonstrate that nano-DSF is also suitable for studying the protein interactions with real food systems that are rich in variety of PPs such as green tea, coffee and cocoa.

### 2.1. Method Development and Validation

In the first set of experiments, we optimized the buffer and concentration of selected proteins, obtained temperature-dependent DSF curves scanning from 20–90 °C and back monitored the change in the 330 nm to 350 nm fluorescence ratio F_350_/F_330_. A change in this ratio indicates denaturation of the protein by conformational change. An increase in the 350 nm fluorescence indicates an increase in Trp moieties exposed to a more polar aqueous environment. To better visualize the curves, a first derivative of the temperature fluorescence curve was obtained. The curves were shown to be fully reversible with all the proteins refolding to their original native states in the absence of a polyphenol, as seen in the DSF curves on cooling. These experiments demonstrate the suitability of the proteins selected, possessing a sufficient number of tryptophane moieties, allowing monitoring of conformational change. The number of tryptophans per protein range from as low as four in ovalbumin to 40 in ACE-2. Selected examples are shown in Figure 2 and in the Appendix A. In all cases, we observed a change in the fluorescent response clearly indicating that, at chosen concentrations, conformational changes can be monitored.

In a second experiment, we added selected polyphenols to the protein solution and again measured temperature-dependent DSF curves scanning from 20 to 90 °C and back. Two observations were made. For some polyphenols, in addition, no changes in DSF curves were observed (e.g., for caffeic acid); whereas for other polyphenols, the F_350_/F_330_ ratio changed, indicating a conformational change upon interactions between the polyphenol and protein. We interpret this fluorescence change as a non-covalent binding event of the PP to the protein, inducing conformational changes leading to alterations of the Trp environment. The principle is graphically illustrated in Figure 2. Secondly, upon the addition of the polyphenol to the protein, some DSF curves were not reversible upon cooling, indicating that PP–protein interactions lead to a misfolding of a PP–protein complex upon binding of the polyphenol to the protein.

Most importantly, for some polyphenol–protein mixtures, identical DSF temperature curves were obtained if compared to the pure protein. This indicates that the addition of a polyphenol gratifyingly does not affect the fluorescence measurement, although all polyphenols show overlap in the UV absorption curve with the tryptophane absorption at 280 nm, and, e.g., hydroxycinnametes could in theory quench Trp fluorescence with absorption maxima at 320 nm, causing false-positive responses. This was evidently not the case at the concentrations employed. Hence, these data serve as both positive and negative control experiments indicating the validity of DSF measurements. Thus, the inner filter effect by reabsorption of the tryptophane fluorescence emission by the polyphenols can be neglected.

### 2.2. Quantification Experiments

In a next round of experiments, we measured nano-DSF curves at two temperatures, 25 °C and 37 °C, at a constant protein concentration of 2.5–5 µM with a variation in polyphenol concentration varying from 0.001 µM to 1 mM. Nano-DSF experiments do not provide information on possible precipitation. At the highest polyphenol concentration, a light-scattering experiment using laser diffraction as a complementary technique was carried out to demonstrate the absence of insoluble complex particle formation at the protein concentrations employed. For example, for lipase, no precipitation was observed at the 2.5 µM concentration, whereas light scattering was observed at 5 µM.

From these binding curves, in theory, quantitative binding data can be derived, in case the binding stoichiometry is known. However, experimentally accurate binding stoichiometry determination requires protein/PP concentrations leading to precipitation of protein–polyphenol insoluble complexes and is therefore impossible with current analytical methods. However, assuming a 1:1 binding stoichiometry curve fitting result, the binding constants shown in Table 1 as representative examples. In particular, the binding constants of 5-CQA to serum bovine albumin (BSA) and PGG obtained here are in good agreement with those reported in the literature [25,26], hence serving as an independent validation of the method.

Analytical parameters were determined in this step for three pairs of PP–protein complexes. Following eight analytical replicate measurements at three different α-PGG and 5-CQA concentrations with BSA and caftaric acid with amylase, the RSD of the F_350_/F_330_ value was determined to be 2.1–3.2%. RSD values for intra-day reproducibility and intra-operator reproducibility were determined to be 2.2% and 2.6%, respectively, from eight technical and analytical replicates each. LOD and LOQ values depend on the protein under investigation, depending crucially on the number of trp residues involved in conformational change. For pragmatic reasons, we recommend to ascertain the highest protein concentration that does not lead to precipitation, that allows concomitant observation of a F_350_/F_330_ change. From our data, the lowest LOD observed was for ACE-2 with a total of 40 Trp residues at 0.001 µM, but this was 0.1 µM for ovalbumin with only four tryptophane residues.

Using the selection of proteins mentioned above and polyphenols, a total of 57 combinations of polyphenols and proteins were investigated in this way, which already indicated interactions in the previous experiments. The resulting curves can be classified in three distinct shapes, as shown in Figure 3. Firstly, a typical asymptotic ascending curve is obtained with an increase in the F_350_/F_330_ ratio with increasing PP concentration. This type of curve indicates the binding of a polyphenol to a protein with associated conformational change exposing trp moieties to an aqueous environment. In subsequent discussions, we refer to this behavior as unfolding of the protein.

A second type of curve shows an asymptotic descending curve with a decrease in the F_350_/F_330_ ratio with increasing PP concentrations, pointing toward the binding of a polyphenol to a protein with associated conformational change exposing trp moieties to a more lipophilic protein environment. Hence, conformational changes induced upon binding can alter the trp environment in both directions. In subsequent discussions, we refer to this behavior as the folding of a protein.

The last type of curve shows a straight line corresponding to no conformational change affecting Trp moieties and therefore no interactions. As exceptions, on three occasions, curves with plateaus were observed with plateaus and a stepwise change in the F_350_/F_330_ ratio. This behavior was observed, for example, for EGCG binding to casein, ACE-1 or lipase. We interpret this behavior as stepwise binding of several EGCG ligands to the protein to several distinct binding sites, with each binding event leading to a change in conformation and Trp environment. Each binding event possesses a distinct binding constant and is slow on the experimental time scale. A double sigmoidal two-step binding pattern was previously reported by Sinisi and Forzato for the binding of 5-CQA to BSA in fluorescence quenching titrations [26].

Table 2 summarizes the qualitative binding behavior observed. Within the table, we qualitatively classify the binding behavior observed and state the experimental F_350_/F_330_ ratio at a concentration of 1 mM of polyphenol at a constant protein by weight concentration. A value larger than the F_350_/F_330_ ratio of the protein solution in absence of the PP indicates unfolding and a value smaller than this value indicates folding of the protein in the presence of the PP. Experimental data as binding curves are shown in Figure 2 and Figure 3. Within the figures, we show multiple fluorescence concentration curves organized by the polyphenol assayed with a selection of proteins and, additionally, multiple polyphenols interacting with a single selected protein. From the data, it can be concluded that polyphenols of very different chemical structures show interaction with a selected protein, for example, the digestive protease chemotrypsin. Within the figure, four out of six selected polyphenols, namely 5-CQA, caffeic acid, EGCG and epicatechin, show an interaction, whereas glycosidic-bound polyphenols show no interaction. Similarly, a single polyphenol, for example, 5-CQA, shows interaction with multiple proteins including pepsin, chemotrypsin, BSA, OVA and amylase.

In the last part of this study, we decided to investigate a series of further enzymes with respect to phenol binding as well as interactions between real foods composed of a mixture of several constituents. As further enzymes, we decided to investigate typical digestive enzymes that are expressed in the human digestive tract. Interaction and ultimately inhibition of these enzymes might lead to reduced levels of enzymatic digestion and provide a rationale for the effects of several polyphenols against diabetes or metabolic syndrome [27]. Griffiths pointed out as early as 1986 that polyphenols precipitate digestive enzymes leading to a depletion of nutrients [28]. These enzymes investigated included lipase, [29] pepsin, trypsin [22] and amylase [30], covering lipid, protein and carbohydrate metabolism. In a recent review, data on in vitro interaction between PPs and digestive enzymes studied by circular dichroism and calorimetry gives an alternative approach to our nano-DSF strategy [31]. Several previous studies have already demonstrated the inhibitory effects of PPs on amylase, [30] lipase [29] or human proteases [22]. Additionally, we investigated the interaction between PPs and mucin, a glycoprotein secreted by epithelial goblet cells, forming a protective layer in the small intestine. Here, no data on PP–mucin interactions have been published. Pantelis suggested that galloyl esters would cross-link mucin [32]. A modulation of mucin structure by PPs might have an impact on gut health. Lastly, we added ACE-1 (angiotensin-converting enzyme 1) to the list of proteins investigated due to its relevance in blood pressure control and being a membrane-bound enzyme [33].

From the experimental curves (Figure 4), it becomes apparent that a given PP interacts in the same way with any given protein tested, e.g., 5-CQA or PGG always result in a folding of the proteins, whereas EGCG or epicatechin always result in an unfolding of the proteins.

### 2.3. Screening of Dietary Material

In terms of real dietary extracts, we investigated green tea, green coffee and cocoa extract. Black tea and roasted coffee displayed a strong background fluorescence, impeding measurements at higher concentrations. For all dietary materials, total anti-oxidant capacity was determined using the FRAP method and gallic acid equivalents were used as a proxy for PP concentrations in quantitative experiments. Additional LC–MS data in the negative ion mode were obtained to validate the chemical composition present in cocoa [34] (see Appendix A), coffee [35] and tea [36]. These correspond in all cases to those previously reported. In all cases, changes in Trp fluorescence were observed, indicating the binding of PPs with subsequent conformational change. It can be assumed that, e.g., in green and roasted coffee with 1.3 mM of 5-CQA and 0.3 mM of 3,4 di-CQA, PPs are present and exerting an effect comparable to the authentic standard material [37,38,39]. Figure 5 shows representative examples of the nano-DSF titration curve for two representative proteins, trypsin and pepsin.

## 3. Discussion

In this study, we have developed and introduced a simple and rapid method, facilitating the investigation of a decade-old problem in polyphenol chemistry, namely the experimental measurement of polyphenol–protein binding and interaction avoiding precipitation. In principle, the method allows the determination of the binding constant at known binding stoichiometries. In the absence of this parameter, qualitative studies are possible. The data clearly show that the majority of PPs bind to most proteins with binding constants in the upper µM range.

Most polyphenols bind to typical human digestive enzymes, confirming previous studies on enzyme inhibition. Any binding at the active site might directly result in enzyme inhibition, or more likely, non-selective binding associated with conformational change might lead to a loss of catalytic function of the enzyme [40]. The binding process occurs in most cases at dietary or even sub-dietary concentrations of the PPs studied. For example, a cup of coffee contains 1.5–2.2 mM 5-CQA a cup of green tea contains 0.6–0.9 mM EGCG and a bar of dark liquified chocolate contains 2.4 mM of epicatechin (data from phenol explorer data, www.phenol-explorer.eu). As is demonstrated with the ACE receptor example, the same observations hold for pharmaceutically relevant protein receptors. Finally, we demonstrated a significant affinity of dietary polyphenols to mucin, a protein relevant to lining the epithelial cells in the lower digestive tract. We hope that the new analytical method introduced will open up avenues allowing a full understanding of this highly relevant class of dietary constituents and their interaction with biological targets relevant to human health.

The promiscuity observed indicates a lack of selectivity of dietary PPs towards putative biological targets. A given polyphenol does not bind selectively to a single target, which is a requirement for the drug development of any pharmaceutical. It binds to several targets. Similarly, a given protein biological target shows affinity to several polyphenolic compounds. Hence, the lack of selectivity, which we refer to as promiscuity, occurs on three levels, as summarized in Figure 6. Previous studies have demonstrated that a Hill coefficient larger than one adds a third level of promiscuity with multiple polyphenol molecules binding to a single protein target.

It might be worth speculating about the relevance of these observations. As a first line of thought, we like to remind the reader of the role of polyphenols in evolution. Research by Thornton has shown that in evolution, enzymes and receptors develop in the absence of their future ligands or substrates [41]. Instead, it was suggested that ancestral precursors with an affinity to proteins play a crucial role in evolutionary protein selection [42]. We suggest here that polyphenols, an integral part of any herbivore’s diet, could play such a role in the evolution of human and animal proteins, providing an initial weak ligand that allows evolutionary changes and the development of protein structure.

Secondly, polyphenols occur in nature as part of our diet as a mixture of many compounds. On some occasions, for example, in black tea chemistry, tens of thousands of different PPs are formed during fermentation, each at low concentrations. Promiscuity in terms of many different compounds binding to the same target might result in an additive effect, resulting in improved health parameters.

Most importantly, polyphenols are beneficial for human health, which is beyond doubt by scientific consensus. Here, we demonstrate multiple interactions and a lack of selectivity in binding on multiple levels. This can be viewed as a small perturbation of a biological system after polyphenol intake. We suggest that such multiple small perturbations overall result in an improved function of a human as a biological system. The effects of such small perturbations have been described on several occasions in the mathematical discipline of stability theory. For example, Guo [43] could show how multiple small perturbations in a system restore stability at equilibrium. Similar concepts using Lyapunov stabilities have been applied to biological systems by Blanchini [44].

## 4. Materials and Methods

### 4.1. Chemicals and Reagents and Sample Preparation

Proteins and reagents were purchased from Sigma–Aldrich, and polyphenols were purchased from Phytolab (Vestenbergsgreuth, Germany). PGG was isolated as described previously [45].

Phenol and protein stock solutions were prepared in water and stored at 4 °C to prevent polyphenol oxidation and protein proteolysis. Vortex and sonication were used to assist dissolution. Before use, stock solutions were diluted to the desired concentrations at room temperature of 20 °C.

For phenol–protein interactions, the protein concentrations were kept constant (0.05 or 0.25 g/L) while the phenol concentrations were varied from 0.05 to 1.0 g/L. All samples were analyzed by nano-differential scanning fluorimetry (nano-DSF). Protein solutions were prepared in a phosphate buffer at pH 6.8 (all proteins were investigated, only pepsin was at pH 2.5). Protein concentrations were realized by serial dilution series. Polyphenol stock solutions were prepared in a phosphate buffer at pH 6.8, and following serial dilution, they were mixed with protein solutions. The solutions were allowed to stand for 10 min and were subsequently transferred to a glass capillary and subjected to measurements.

Plant extracts were prepared as described previously and added without pH adjustment to the protein solutions [46,47].

### 4.2. Nano-DSF Measurements

The thermal unfolding profiles of polyphenol–protein mixtures were measured using Nano-DSF (Prometheus NT.48, NanoTemper Technologies, Munich, Germany). Nano-DSF records the fluorescence intensity of tryptophan and tyrosine residues in proteins at emission wavelengths of λ = 330 and 350 nm depending on the composition of a protein solution and scans the solution at different temperatures. The emission wavelength of amino acids is highly sensitive towards their environment, allowing monitoring of their micro-environment and hence the conformational state of the protein itself. Thus, the specific fluorescence ratios (F350/F330) of the unfolded and folded states can be directly associated with the respective fraction of unfolded proteins at any point during the unfolding process [30]. Nano-DSF can measure 48 samples in parallel with a capillary loading volume of only 10 μL. All measurements were performed at both increasing and decreasing temperatures, undergoing folding and unfolding phases with a rate of 1 °C/min (from 20 to 95 °C and vice versa), aiming to check whether the phenol binding was stable when the temperature decreased. In addition to this, the recovered samples were stored at 4 °C and measurements were repeated after 48 h in the same way. The protein–polyphenol-complex molecule seemed to remain stable.

### 4.3. Anti-Oxidant Essay (FRAP) for Green Tea, Cocoa and Coffee Extract

Solution 1 was prepared by dissolving 15.92 mg of TPTZ in 5 mL of 40 mM HCL, and Solution 2 consisted of 27.44 mg FeCl3 hexahydrate in 5 mL of deionised water. The final FRAP reagent was then made prior to use by mixing 4 mL of Solution 1, 4 mL of Solution 2 and 40 mL of acetate buffer (pH = 3.6). The standard FRAP assay was modified by only using gallic acid as a FRAP standard. The FRAP assay was carried out in triplicate, and prepared in a 96-microwell plate, in which 10 μL of the three varying concentrations (1.0 g/L, 0.5 g/L and 0.1 g/L) of green tea extract, green coffee extract, cinnamon extract and acticoa, followed by 200 μL of the FRAP reagent, were added. The well was then incubated in the dark for 10 min before photometric measurements were taken at a wavelength of 593 nm

### 4.4. Curve Fitting

Data obtained from the binding studies were used for the calculations of the binding parameters (binding constant Ka and number of binding sites n). Considering the interactions between proteins and polyphenols as protein–ligand binding, then the thermodynamic equilibrium can be described as follows:*Protein (P) + Polyphenol (L) ↔ Protein − Polyphenol Complex (PL)*

The best fit values for binding parameters were achieved by applying nonlinear least squares regression using the Solver tool in Microsoft Excel. The following equation was applied [48]:$$[PL]=\frac{1}{2}\left({\left[P\right]}_{0}+{\left[L\right]}_{0}+\frac{1}{K_{a}}+\sqrt{{\left({\left[P\right]}_{0}+{\left[L\right]}_{0}+\frac{1}{K_{a}}\right)}^{2}-4{\left[P\right]}_{0}{\left[L\right]}_{0}}\right)$$
where [*PL*] is the concentration of the protein–polyphenol complex at equilibrium, [*P*]_0_ and [*L*]_0_ are the initial concentrations of protein and polyphenol, respectively, and Ka is the binding (association) constant.

## Figures and Tables

**Figure 1 molecules-30-00965-f001:**
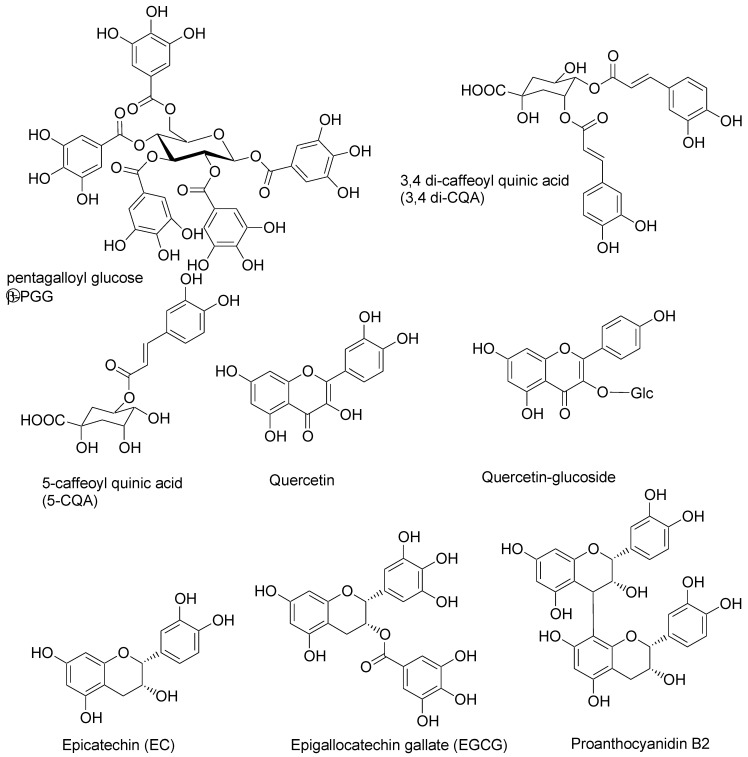
Chemical structures of relevant dietary polyphenols studied.

**Figure 2 molecules-30-00965-f002:**
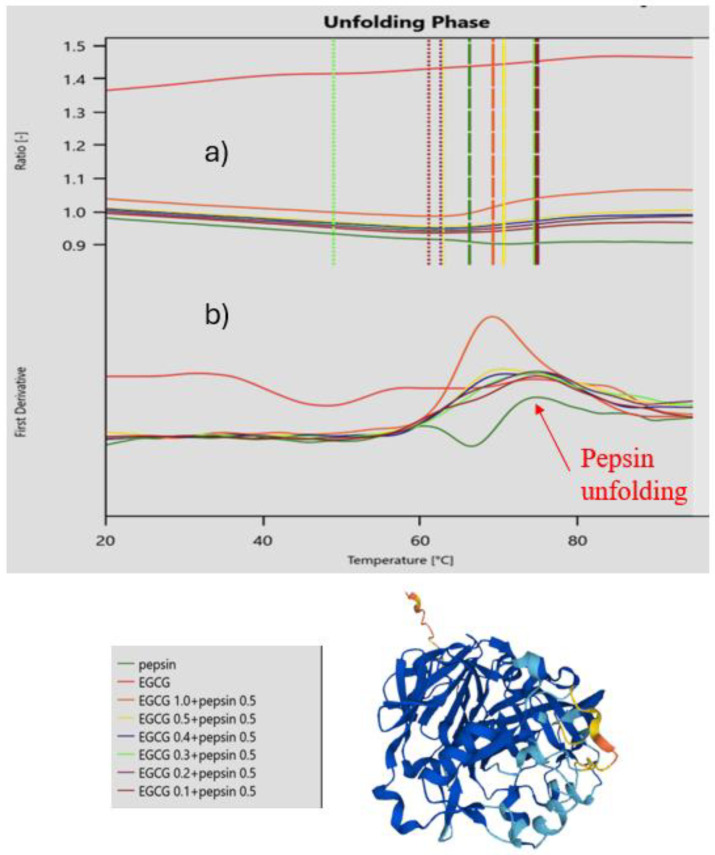
Nano-DSF curves showing change in F_350_/F_330_ ratio upon heating of pepsin in the absence and presence of varied concentrations of EGCG (decimals indicate weight ratio of pepsin to EGCG). (**a**) Direct measurement of F_350_/F_330_ ratio versus temperature; (**b**) first derivative of F_350_/F_330_ ratio versus temperature. The curves indicate a change in unfolding processes in the presence of EGCG as a consequence of EGCS binding to the protein. The AlphaFold structure of pepsin is shown in the diagram.

**Figure 3 molecules-30-00965-f003:**
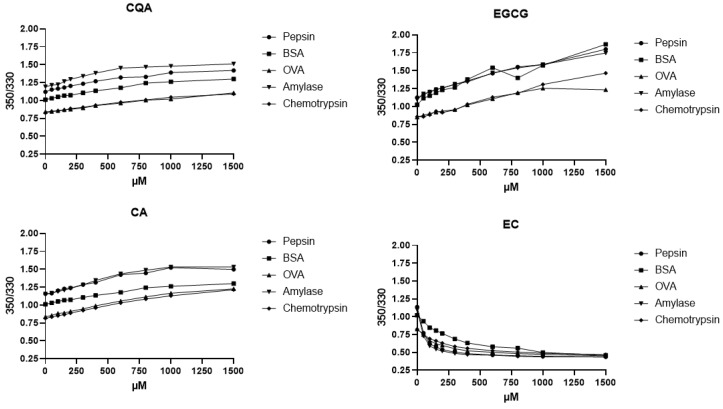
F_350_/F_330_ ratio concentration curves of four selected polyphenols (5-caffeoyl quinic acid (CQA), epi-gallocatechin gallate (EGCG), caftaric acid (CA) and epi-catechin (EC) against five selected proteins at constant protein concentration and varied polyphenol concentration (error bars small; experiments carried out in triplicate).

**Figure 4 molecules-30-00965-f004:**
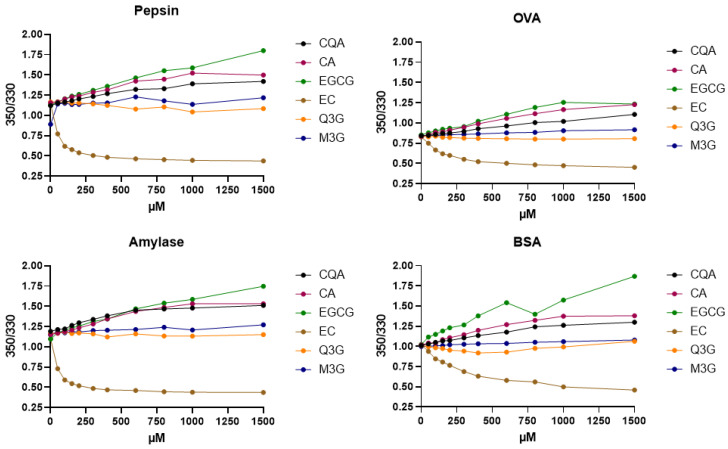
F_350_/F_330_ ratio concentration curves of four selected proteins (including ovalbumin (OVA), bovine serum albumin (BSA)) against six selected polyphenols (5-caffeoyl quinic acid (CQA), epi-gallocatechin gallate (EGCG), caftaric acid (CA), epi-catechin (EC), quercetin-3-glucoside (Q3G) and malvidin-3-glucoside (M3G)) at constant protein concentration and varied polyphenol concentration (experiments in triplicate).

**Figure 5 molecules-30-00965-f005:**
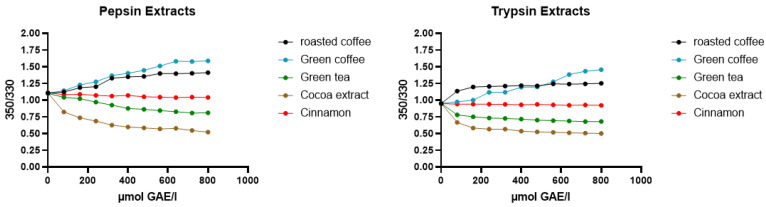
F_350_/F_330_ ratio concentration curves of two selected proteins against five selected dietary aqueous extracts at constant protein concentration and varied extract concentration. Extract concentration given as gallic acid equivalents (GAE) determined by FRAP assay (experiments carried out in triplicate).

**Figure 6 molecules-30-00965-f006:**
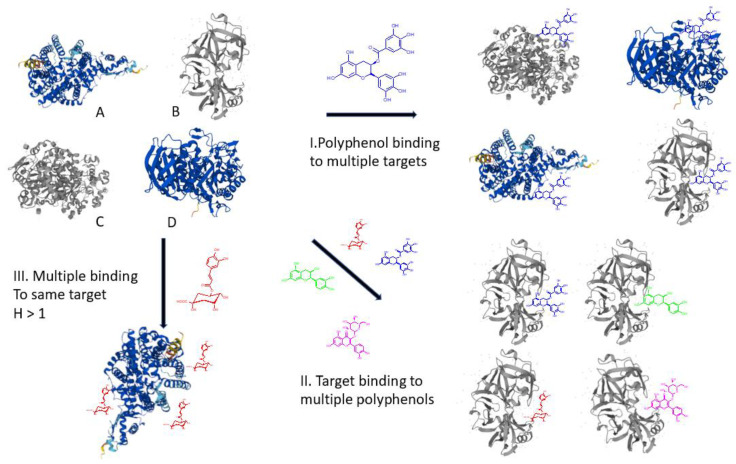
Schematic diagram of polyphenols interacting with proteins (AlphaFold protein structures of human lipase (**A**), pepsin (**B**); amylase (**C**) and ACE-1 (**D**)) obtained from Uniprot as AlphaFold models (www.uniprot.com (accessed on 1 April 2024)).

**Table 1 molecules-30-00965-t001:** Quantitative binding data for polyphenol–protein interactions using curve fitting assuming a 1:1 binding stoichiometry.

Protein	Polyphenol	Temperature °C	K_a_ [M^−1^]	K_a_ [µM]
BSA	Tannic acid	20	29 878	33.5
BSA	Tannic acid	37	26 269	38.01
BSA	β-PGG	20	1 890	529.1
BSA	β-PGG	37	8 492	117.8
BSA	α-PGG	20	13 392	74.7
BSA	α-PGG	37	11 196	89.3
BSA	α-PGG	The literature		80 µM
BSA	5-CQA	20	28 387	35.2
BSA	5-CQA	37	26 917	37.2
BSA	5-CQA	The literature		32 µM
BSA	3,4-diCQA	37	55 601	17.9
Casein	5-CQA	37	21 450	46.6
Casein	3,4-diCQA	37	41 271	24.2
Casein	Epicatechin	37	7 1616	139.6

**Table 2 molecules-30-00965-t002:** Summary of nano-differential fluorimetry measurements. Table is showing the F350/F330 ratio of free protein in solution at 1 mg/mL concentration (first row) and F350/F330 value at 1 mmol (highest concentration) of polyphenol employed or 800 µmol GAE equivalent for all extracts. A value different from the free protein indicates a conformational change upon binding. Values smaller than free protein indicate a folding process; a larger value, if compared to the free protein, indicates an unfolding event; and a value unchanged within experimental error indicates no interaction. ND indicates data not determined or opaque solution. * The superscript indicates the concentration.

	BSA	Ovalbumin	Mucin	α-Amylase	Lipase	Chemotrypsin	Pepsin	Trypsin	ACE-1
Free protein F_350_/F_330_	1.02	0.84	1.00	1.14	1.05	0.83	1.10	0.95	1.01
MW Protein in kDa	66.5	44.3	640	51.5	120	25.0	35.0	23.8	138
Molarity in assay [µM]	5	5	2.5	5	2.5	5	5	5	2.5
**Polyphenol**									
5-CQA	1.26	1.02	1.40	1.48	1.42	1.05	1.39	ND	1.28
3,4 di CQA	ND	ND	ND	ND	1.50	ND	ND	ND	ND
Quercetin	ND	ND	ND	ND	1.04	ND	ND	ND	0.99
EGCG	1.57	1.25	1.15	1,59	0.88	1.31	1.59	1.61	0.81
Epicatechin	0.50	0.47	0.51	0.44	0.78	0.49	0.45	ND	ND
Procyanidin B2	ND	ND	ND	ND	0.92	ND	ND	ND	ND
Caftaric acid	1.374	1.17	1.38	1.53	ND	1.13	1.52	1.45	ND
Cyanidin-3-glucoside	ND	ND	1.02*^800µM^	ND	ND	ND	ND	ND	ND
Malvidin-3-glucoside	1.06	0.90	1.05	1.21	ND	0.87	1.14	ND	ND
Quercetin-3-glucoside	0.99	0.80	0.94	1.13	ND	0.79	1.04	ND	ND
α-PGG	1.91^*530µM^	1.74^*530µM^	ND	ND	1.38	1.22	1.35^*530µM^	ND	ND
β-PGG	1.60^*530µM^	1.50^*530µM^	ND	ND	ND	ND	1.09^*530µM^	ND	ND
Tannic acid	1.56*^300µM^	1.38*^150µM^	ND	ND	1.24	1.41	1.26*^300µM^	ND	ND
Caffeic acid	ND	ND	ND	ND	1.15	0.85	ND	ND	ND
Gallic acid	1.07^*1176µM^	1.07^*1176µM^	ND	ND	1.07	0.81	1.08^*1176µM^	ND	ND
Green Tea	0.84	0.66	0.98^*1mg/mL^	ND	0.87	0.78	0.81^*0.8mmol GAE/l^	0.68	0.65
Black tea	0.91	0.67	ND	ND	ND	0.74	0.72^800µmol GAE/l^	0.75	0.48
Cocoa extract	1.18	0.82	0.78^*1mg/mL^	ND	0.98	0.82	0.52^*800µmol GAE/l^	0.50^*800µmol^	No raw data
Green coffee extract	1.45	1.41	ND	ND	1.32	1.34	1.59^*800µmol GAE/l^	1.46^*800µmol^	1.55
Roasted coffee extract	1.37	ND	1.05^*1mg/mL^	ND	ND	1.28	1.41^*800µmol GAE/l^	1.25^*800µmol^	ND

## Data Availability

Data included in Appendix A and available on request.

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
