# Peer review of "Promiscuity in Polyphenol–Protein Interactions—Monitoring Protein Conformational Change upon Polyphenol–Protein Binding by Nano-Differential Fluorimetry (Nano-DSF)"

_molecules, 2025, doi:10.3390/molecules30040965_

Round 1
Reviewer 1 Report
Comments and Suggestions for Authors
The authors test a wide range of proteins for interactions and effects with many polyphenols.
This fluorescence-based technology is attractive allowing use of lower protein concentrations than some other assays, therefore avoiding or reducing the confounding effects of precipitation. Avoiding an additional hydrophobic fluorophore (e.g. ANS or SYBR dyes) with binding assays of these hydrophobic ligands sounds a good choice.
1. Could the authors comment on the risk of inner filter effects? Perhaps the concentrations of reagents and the dimensions of the capillaries reduce this risk, but I would like to read this addressed.
2. The authors suggest one advantage of their nano-DSF strategy is avoiding the precipitation at higher protein concentrations used in other assays. However they used (near-) micromolar protein concentrations in some tests. Would the nano-DSF technology report precipitation if it occurred?
3. The “folding” assessment of EC interaction with all of pepsin, ovalbumin, amylase and BSA (and other proteins in Table 2) is surprising. Some of these are stably-folded, globular proteins – we use them as size markers for gel filtration chromatography and in other assays. They wouldn’t work well in those applications if natively partially-folded. The structures presented in Figures 2 and 6 support these being stably-folded proteins.
Is it more likely that EC has an effect on the environment around Trp residues changing fluorescence without extrapolating this to concluding folding (lines190-192)?
Fluorescence is often an empirical measure of molecular properties with various explanations possible for the changes in spectrum or intensity seen.
4. Could the authors comment on why EC is the only molecule showing “folding” effect? I notice that the structurally similar quercetins have not been comprehensively tested for effects (Table 2).
Minor comments
· Figure 2 looks like a screen-shot. Please re-plot.
· Convert protein concentrations to molarity, in addition to the quoted w/v measures.
· Consider re-plotting graphs, e.g. S2 - S11, using the same y-axis range. Certainly remove redundant decimal places.
· Last three supplementary figures lacking numbers to figure legend.
Author Response
Referee 1
The authors test a wide range of proteins for interactions and effects with many polyphenols.
This fluorescence-based technology is attractive allowing use of lower protein concentrations than some other assays, therefore avoiding or reducing the confounding effects of precipitation. Avoiding an additional hydrophobic fluorophore (e.g. ANS or SYBR dyes) with binding assays of these hydrophobic ligands sounds a good choice.
- Could the authors comment on the risk of inner filter effects? Perhaps the concentrations of reagents and the dimensions of the capillaries reduce this risk, but I would like to read this addressed.
A comment has been added. We were worried about fluorescent quenching of the phenolics when starting the experiments. However the nano DSF signal is stable and independent of phenol concentration and does change when switching the protein. Hence we concluded that no filter effect occurs.
- The authors suggest one advantage of their nano-DSF strategy is avoiding the precipitation at higher protein concentrations used in other assays. However they used (near-) micromolar protein concentrations in some tests. Would the nano-DSF technology report precipitation if it occurred?
Nano DSF would not report on precipitation. We mentioned in the paper that we spot checked some samples at high protein concentrations, using dynamic light scattering with a Malvern Mastersizer instrument from the lab of my colleague Wintherhalter. Here no evidence of particle formation was evident with light scattering being an appropriate technique to monitor insoluble particle formation. The protein concentrations stated are in mg/ml resulting in low molarities. Concentration data have nbeen added to clarify matters.
- The “folding” assessment of EC interaction with all of pepsin, ovalbumin, amylase and BSA (and other proteins in Table 2) is surprising. Some of these are stably-folded, globular proteins – we use them as size markers for gel filtration chromatography and in other assays. They wouldn’t work well in those applications if natively partially-folded. The structures presented in Figures 2 and 6 support these being stably-folded proteins.
Is it more likely that EC has an effect on the environment around Trp residues changing fluorescence without extrapolating this to concluding folding (lines190-192)?
Fluorescence is often an empirical measure of molecular properties with various explanations possible for the changes in spectrum or intensity seen.
I completely agree with the referee and I do not have a good explanation. On a general basis from our data globular proteins tend to interact less with polyphenols. On the proteins all Trp are in different environments and I would suggest that EC binds to special Trp moiety depending on their molecular context. In the absence of any clear data to support this notion we decided to keep the data uncommented.
- Could the authors comment on why EC is the only molecule showing “folding” effect? I notice that the structurally similar quercetins have not been comprehensively tested for effects (Table 2).
See above. I have no obvious explanation but we keep the observation in mind and see whether at a later stage we can collect data that would allow for a rationale.
Minor comments
- Figure 2 looks like a screen-shot. Please re-plot.
It is the actual export of the instrument software. I prefer to stay close to original data so I decided to keep it as it is.
- Convert protein concentrations to molarity, in addition to the quoted w/v measures.
Done
- Consider re-plotting graphs, e.g. S2 - S11, using the same y-axis range. Certainly remove redundant decimal places.
Changed
- Last three supplementary figures lacking numbers to figure legend.
Changed
Referee 2
General Comments
This study employs nano Differential Scanning Fluorimetry (NanoDSF) to investigate polyphenol-protein interactions. NanoDSF is a rapid and effective tool for assessing protein stability and conformational changes. This study represents an innovative application of NanoDSF to measure and understand the binding behavior of various polyphenols with a diverse range of proteins. Overall, the research is scientifically compelling and impactful. The experiments are well-designed, and the data provided sufficiently support the conclusions. Additionally, the manuscript is well-organized and well-prepared. However, a few revisions could further enhance the completeness and robustness of the scientific narrative.
One key finding of this study is the dual-level promiscuity observed in relation to both protein targets and polyphenol structures. While this is a significant contribution, the conclusion could be further strengthened by incorporating complementary experiments or investigations. For instance, molecular docking simulation studies could be beneficial in this context. Such simulations may help identify binding sites, elucidate the mechanisms of interaction, and, importantly, evaluate the observed promiscuity in binding behavior.
Many thanks for the referees suggestion. I completely agree that additional studies in the future of the type suggested are worth undertaking. Unfortunately at the current state additional computational work is outside the scope of the current submitted experimental paper and would require expert collaborators.
Specific Comments
- Line 110: There is a typo in "thistechnique"; it should be corrected to "this technique."
Changed
- Line 172: Figure 2 presents curve images directly from the instrument, but the quality is relatively low. Additionally, the figure legend does not provide sufficient explanation, nor does the main text clarify the figure adequately. For example, the dashed line in Figure 2A should be described to make the figure more comprehensible, particularly for readers who are not experts in this field.
Indeed it is an export from the manufacturer software. I added some further explanations to the figure shown.
- Line 196: It would improve clarity if the Limit of Detection (LOD) and Limit of Quantification (LOQ), along with the methods used to determine them, were provided for each group. Additionally, specifying the quantification or linear range would enhance the understanding of the assay's quantification capabilities.
In the interaction assays the resulting binding curves are not linear but follow the equation stated. Hence assessing linearity would not necessarily make sense and conventional definition of LOQ values defined by the range of linearity of the calibration curve would not really fit in my view.
The limit of detection would vary for each protein investigated depending on the number of Trp residues that undergo a change of the fluorescent response. We checked in details for proteins with low Trp numbers whether at the concentrations employed a change of signal could be reliably detected. This would be a surrogate number for an LOD value.
- Line 261: In Figures 3, 4, and 5, it would be helpful to clarify the number of experimental replicates performed. Including this information in the figure legends will ensure that the results' reliability and reproducibility are transparent.
Information added
Recommendation
Overall, this study is interesting, innovative, and impactful. However, considering the issues outlined above, I recommend major revisions to improve the manuscript further.
Reviewer 2 Report
Comments and Suggestions for Authors
General Comments
This study employs nano Differential Scanning Fluorimetry (NanoDSF) to investigate polyphenol-protein interactions. NanoDSF is a rapid and effective tool for assessing protein stability and conformational changes. This study represents an innovative application of NanoDSF to measure and understand the binding behavior of various polyphenols with a diverse range of proteins. Overall, the research is scientifically compelling and impactful. The experiments are well-designed, and the data provided sufficiently support the conclusions. Additionally, the manuscript is well-organized and well-prepared. However, a few revisions could further enhance the completeness and robustness of the scientific narrative.
One key finding of this study is the dual-level promiscuity observed in relation to both protein targets and polyphenol structures. While this is a significant contribution, the conclusion could be further strengthened by incorporating complementary experiments or investigations. For instance, molecular docking simulation studies could be beneficial in this context. Such simulations may help identify binding sites, elucidate the mechanisms of interaction, and, importantly, evaluate the observed promiscuity in binding behavior.
Specific Comments
1. Line 110: There is a typo in "thistechnique"; it should be corrected to "this technique."
2. Line 172: Figure 2 presents curve images directly from the instrument, but the quality is relatively low. Additionally, the figure legend does not provide sufficient explanation, nor does the main text clarify the figure adequately. For example, the dashed line in Figure 2A should be described to make the figure more comprehensible, particularly for readers who are not experts in this field.
3. Line 196: It would improve clarity if the Limit of Detection (LOD) and Limit of Quantification (LOQ), along with the methods used to determine them, were provided for each group. Additionally, specifying the quantification or linear range would enhance the understanding of the assay's quantification capabilities.
4. Line 261: In Figures 3, 4, and 5, it would be helpful to clarify the number of experimental replicates performed. Including this information in the figure legends will ensure that the results' reliability and reproducibility are transparent.
Recommendation
Overall, this study is interesting, innovative, and impactful. However, considering the issues outlined above, I recommend major revisions to improve the manuscript further.
Author Response
Referee 2
General Comments
This study employs nano Differential Scanning Fluorimetry (NanoDSF) to investigate polyphenol-protein interactions. NanoDSF is a rapid and effective tool for assessing protein stability and conformational changes. This study represents an innovative application of NanoDSF to measure and understand the binding behavior of various polyphenols with a diverse range of proteins. Overall, the research is scientifically compelling and impactful. The experiments are well-designed, and the data provided sufficiently support the conclusions. Additionally, the manuscript is well-organized and well-prepared. However, a few revisions could further enhance the completeness and robustness of the scientific narrative.
One key finding of this study is the dual-level promiscuity observed in relation to both protein targets and polyphenol structures. While this is a significant contribution, the conclusion could be further strengthened by incorporating complementary experiments or investigations. For instance, molecular docking simulation studies could be beneficial in this context. Such simulations may help identify binding sites, elucidate the mechanisms of interaction, and, importantly, evaluate the observed promiscuity in binding behavior.
Many thanks for the referees suggestion. I completely agree that additional studies in the future of the type suggested are worth undertaking. Unfortunately at the current state additional computational work is outside the scope of the current submitted experimental paper and would require expert collaborators.
Specific Comments
- Line 110: There is a typo in "thistechnique"; it should be corrected to "this technique."
Changed
- Line 172: Figure 2 presents curve images directly from the instrument, but the quality is relatively low. Additionally, the figure legend does not provide sufficient explanation, nor does the main text clarify the figure adequately. For example, the dashed line in Figure 2A should be described to make the figure more comprehensible, particularly for readers who are not experts in this field.
Indeed it is an export from the manufacturer software. I added some further explanations to the figure shown.
- Line 196: It would improve clarity if the Limit of Detection (LOD) and Limit of Quantification (LOQ), along with the methods used to determine them, were provided for each group. Additionally, specifying the quantification or linear range would enhance the understanding of the assay's quantification capabilities.
In the interaction assays the resulting binding curves are not linear but follow the equation stated. Hence assessing linearity would not necessarily make sense and conventional definition of LOQ values defined by the range of linearity of the calibration curve would not really fit in my view.
The limit of detection would vary for each protein investigated depending on the number of Trp residues that undergo a change of the fluorescent response. We checked in details for proteins with low Trp numbers whether at the concentrations employed a change of signal could be reliably detected. This would be a surrogate number for an LOD value.
- Line 261: In Figures 3, 4, and 5, it would be helpful to clarify the number of experimental replicates performed. Including this information in the figure legends will ensure that the results' reliability and reproducibility are transparent.
Information added
Recommendation
Overall, this study is interesting, innovative, and impactful. However, considering the issues outlined above, I recommend major revisions to improve the manuscript further.
Round 2
Reviewer 1 Report
Comments and Suggestions for Authors
Some small typos.
line 147 change "in" to "an".
Wherever used "tryptophane" to "tryptophan"
Reviewer 2 Report
Comments and Suggestions for Authors
Revision received and accepted.